# “SHIELDing” Our Educators: Comprehensive Coping Strategies for Teacher Occupational Well-Being

**DOI:** 10.3390/bs14100918

**Published:** 2024-10-09

**Authors:** Joy C. Nwoko, Emma Anderson, Oyelola A. Adegboye, Aduli E. O. Malau-Aduli, Bunmi S. Malau-Aduli

**Affiliations:** 1College of Medicine and Dentistry, James Cook University, Townsville, QLD 4811, Australia; 2College of Public Health, Medical and Veterinary Sciences, James Cook University, Townsville, QLD 4811, Australia; oyelola.adegboye@menzies.edu.au; 3Menzies School of Health Research, Charles Darwin University, Casuarina, NT 0811, Australia; 4School of Environment and Life Sciences, The University of Newcastle, Callaghan, NSW 2308, Australia; aduli.malauaduli@newcastle.edu.au; 5School of Medicine and Public Health, The University of Newcastle, Callaghan, NSW 2308, Australia

**Keywords:** coping strategies, stress, teacher well-being, collaboration, social support

## Abstract

Background: Teaching is a physically and mentally challenging profession that demands high emotional involvement, often leading to stress and anxiety. Understanding how teachers cope with these demands is essential for enhancing their well-being and effectiveness. Objectives: This study aimed to (1) investigate personal and school-based well-being initiatives that teachers use for maintaining their occupational well-being, and (2) develop a coping strategy model that enhances teachers’ occupational well-being. Methods: This study utilised a qualitative phenomenological approach to explore the coping strategies of Australian primary school teachers. Results: The twenty-one participants interviewed employed ten diverse coping strategies classified into five personal and five school-based well-being-enabling initiatives. The personal strategies included setting boundaries, exercise and physical health, social support and interactions, mental health and mindfulness, and work–life balance. The school-based initiatives comprised supportive leadership, colleague support, flexibility and autonomy, resource availability, and proactive approaches to address challenges. A novel SHIELD model incorporating Support, Health, Interaction, Empathy, Leadership, and Development strategies was formulated as a holistic coping strategy for enhancing teachers’ occupational well-being. Conclusions: The findings highlight the necessity of a holistic approach to teacher well-being, integrating both personal and institutional support systems. Schools can enhance teacher well-being by fostering a supportive and empathetic culture, providing necessary resources, and encouraging healthy lifestyles. The SHIELD model offers a comprehensive framework for supporting teachers and improving educational outcomes.

## 1. Introduction

Teaching is a profession that comes with numerous expectations and requirements, set not only by the teachers themselves but also by schools, governments, and other stakeholders [1]. Teachers face a multitude of demands that can result in stress and emotional exhaustion, contributing to a health impairment process [2]. Despite teachers significantly influencing students’ holistic learning, progress, and academic performance [3,4,5], the teaching profession has been reported to be very stressful [6], due to its association with enormous stress, fatigue, and burnout [7]. High levels of teacher occupational stress have been documented globally [3,4,6] and within Australia [3,8], which is the focus of this study. The findings suggest that administrative tasks are a greater source of stress for teachers than classroom teaching hours. Australian teachers, like their international counterparts, spend an average of 19 h teaching, 7 h planning, and 5 h marking each week [3,8]. However, Australian teachers work an average of 43 h per week, which is 5 h more than the global average [3,8,9].

Stress in teachers arises when classroom demands exceed their perceived capacity to cope [10]. This stress impacts their performance globally and is one of the main causes of burnout [5]. Consequently, scholars emphasise the need for approaches to help teachers cope with anxiety [11,12]. The experience of stress among teachers depends on the interaction of personality traits, developed skills, guiding values, and the context generating the stress [13]. Higher stress levels are linked to lower self-efficacy, while better coping mechanisms correlate with higher self-efficacy and increased student pro-social behaviour [10]. Managing difficult classroom behaviours, employing varied pedagogical techniques, promoting student engagement [14,15], and dealing with heavy workloads and intense scrutiny contribute to teacher attrition [16,17]. Additionally, teachers face heightened accountability within the school context [18]. Teachers cite workload, poor work–life balance, and the target-driven culture shaped by government initiatives as key reasons for leaving. While many enter the profession for altruistic reasons—such as wanting to make a difference—the reality of teaching, shaped by accountability and performativity pressures, leads to disappointment and early exit [19]. This reflects a disillusionment with the broader teaching context and the pressures of accountability. Thus, maintaining their well-being is crucial for teachers to fulfil their roles effectively [20]. 

Teacher stress is also linked to attrition [21] and the motivation to leave the profession [22]. The overload of non-teaching activities, such as administrative tasks [16,23,24], contributes to stress, leading to mental ill health, burnout, and ultimately exit from the profession [25]. The literature reports that teaching daily poses numerous challenges, causing teachers to experience stressful events and negative emotions like anxiety or anger during class [26], most especially for primary school teachers, who have reported more stress in the literature than other types of teachers [27,28]. A recent literature review [29] clearly indicated that primary school teachers were the most stressed and may face more challenges in managing disruptive behaviour from young children, which can negatively impact their well-being [30,31]. They may require more energy and patience to handle younger children, which can be exhausting and require a high degree of emotional labour, and experience higher levels of workload and stress due to the constant need for attention and supervision of young children [32]. The perception of teaching as stressful may be influenced by coping responses and social support [33].

Teachers need effective coping strategies to maintain their well-being and teaching quality [34,35]. Poor health and coping abilities in teachers are associated with significant stress levels, suggesting that greater coping skills and well-being can lead to more innovative, challenging, and effective teaching, thereby improving educational outcomes [36]. Teacher coping methods are increasingly recognised as key determinants of teacher effectiveness [5]. Coping, defined as the cognitive and behavioural efforts used to manage specific external or internal demands perceived as exceeding one’s resources [37], represents a valuable resource for individuals dealing with stressors, improving their levels of well-being [11]. Previous research has focused on how coping strategies can alleviate stress and promote a higher quality of work life [38,39]. Teachers employ coping strategies to change stressful situations or make them more manageable when they cannot be altered [12].

International studies suggest that most causes of teacher stress are only weakly associated with coping strategies, thus raising concerns about teachers’ mental health and the effectiveness of their coping mechanisms [5]. Coping strategies are critical for physical and psychological well-being when facing challenges and stress [11]. Bermejo-Toro et al. [7], Kim et al. [40], and Nwoko et al. [29] all reported that coping strategies help to improve teacher well-being and sustain teachers when they experience inadequate support [41,42], preventing burnout [43].

Teachers use a variety of coping strategies to maintain their well-being, often combining several approaches rather than relying on a single method [44]. Hamama et al. [45] suggested that both internal and external coping resources, along with self-control, increase teacher well-being. However, without effective coping strategies, teachers are more likely to experience burnout [46], highlighting the urgent need to reduce stress levels in the teaching profession [12]. Psychological constructs such as personality and self-efficacy significantly contribute to teachers’ coping strategies [47]. Selecting appropriate coping strategies can help future teachers view stressful situations as challenges rather than difficulties, facilitating adaptation to the profession’s demands [48]. Greater teacher coping skills and well-being lead to improved educational outcomes [36].

Addressing teacher coping mechanisms is crucial for dealing with stress and burnout, which negatively affect teachers’ commitment to their profession [49]. Teacher commitment is vital to educational outcomes, attracts high public expectations, and increases the burden on teachers in an already stressful profession [46]. In response, teachers in stressful environments have developed resilience strategies to cope with stressors [50]. Few qualitative studies have explored the coping strategies adopted by primary school teachers to support their well-being, and even fewer have asked teachers to describe their coping strategies holistically even though it is vital [41]. Teacher well-being significantly impacts both the school environment and their families [1]. 

Therefore, this study sought to answer the research question “What are the personal and school-based occupational well-being enabling initiatives that teachers utilise to cope with the demands of their job?” The aims of this study were two-fold: (1) to investigate the personal and school-based well-being-enabling initiatives that teachers use to maintain their occupational well-being, and (2) to develop a coping strategy model that enhances teachers’ occupational well-being. This study sought to provide insights into how teachers use their personal and school coping resources to ensure their well-being, expanding upon prior research and filling the key knowledge gap on maintaining teachers’ well-being.

## 2. Methods

### 2.1. Research Design

This study is part of a bigger project that investigated Australian primary school teachers’ occupational well-being that was guided by the OECD teacher well-being framework, which includes four core well-being dimensions (cognitive, subjective, physical/mental, and social) [16]. The present study adopted a phenomenological qualitative research design to explore the personal and school-based well-being-enabling initiatives that positively influence teacher occupational well-being. The approach focuses on understanding and interpreting teachers’ lived experiences within their natural contexts [51,52]. By employing this approach, this study aimed to explore teachers’ personal narratives and experiences, providing a rich, contextual understanding of how coping strategies influence teachers’ well-being [53] within the primary school educational setting. The phenomenological methodology allows for a comprehensive understanding of the ways in which teacher- and school-initiated coping strategies contribute to their well-being, and seeks to uncover the nuanced, deeply personal experiences of teachers. The focus on primary school teachers is justified by the unique challenges they face, including managing young students with diverse developmental needs, creating foundational learning experiences, and balancing heavy workloads with limited resources [23,24,25]. Primary school teachers often play a critical role in shaping children’s early educational experiences, making their well-being essential for effective teaching and long-term student success. Additionally, primary educators are particularly vulnerable to stress and burnout, which underscores the importance of identifying strategies that support their occupational well-being [27,28,29].

### 2.2. Participant Recruitment and Sampling

A combination of purposive and convenience sampling techniques was used to recruit participants for this study. Initially, participants were recruited using purposive sampling, targeted at registered Australian primary school teachers and school leaders across Australia. Recruitment was conducted through educational forums and social media platforms dedicated to educators. Subsequently, snowballing was utilised to increase participant numbers. All potential participants were provided with detailed information about the study’s aims and the voluntary nature of their participation. 

### 2.3. Data Collection

Data were collected through semi-structured in-depth interviews, allowing for the flexibility to probe deeper into participants’ responses and explore emergent themes. The interview guide was developed based on the research question (see Appendix A) and reviewed by experts in educational psychology and qualitative research to ensure comprehensiveness and relevance. Interviews were conducted via phone call or video conferencing at various times based on the participants’ preferences and were recorded with their consent. Each interview lasted approximately 45–60 min, providing ample time to explore the participants’ experiences and perspectives thoroughly. Before the interview, participants were given a brief overview of the study’s purpose and the interview process, and their consent and demographic variables (context and years of teaching experience) were obtained. To ensure confidentiality, participants’ names were replaced with pseudonyms. The interviews continued until data saturation was achieved [54]. Interviews were transcribed verbatim, and participants were offered the chance to review their transcripts to validate the accuracy and interpretation of their experiences.

### 2.4. Data Analysis

The transcribed data were coded and analysed in NVivo (QSR International/Lumivero Pty Ltd., Denver, CO, USA; Version 12 for windows) by three of the co-authors (JCN, EA and BSMA) to ensure the credibility of the results. The data were subjected to inductive thematic analysis following Braun and Clarke’s six-phase framework [55], which is particularly suited to phenomenological studies. The analytical process involved familiarisation with the data, generating initial codes, searching, reviewing, defining and naming themes, and producing the final report. This approach allowed for the systematic organisation and interpretation of the data, ensuring that the findings were grounded in the participants’ experiences. Throughout the analysis, reflexivity was maintained, with researchers documenting their assumptions, biases, and reflections, to ensure the credibility and authenticity of the findings [56]. Peer debriefing and member checking were employed as additional credibility checks to enhance the trustworthiness of the study’s outcomes. Illustrative quotes were reported verbatim to support the study’s findings. Participants were entered into a draw to win one of four AUD 50 gift cards as part of reciprocity back to the participants for their involvement in the study [53]. 

### 2.5. Ethical Approval

The James Cook University Human Research Ethics Committee granted ethics approval (H8638) for this study. Participants were informed about the study’s purpose, the right to withdraw at any time without any consequence, and the measures taken to protect their privacy and personal data. Informed consent was obtained from all participants prior to their participation in the study. All data were anonymised and securely stored to maintain confidentiality.

## 3. Results

A total of 21 Australian primary school teachers comprising classroom teachers (n = 11), specialist teachers (n = 4), learning support teachers (n = 2), and school leaders (n = 4) participated in this study. Most of the teachers were female (95.2%) and worked in public schools (57.1%). The participants were from five Australian states—Australian Capital Territory (9.5%), New South Wales (23.8%), Northern Territory (19%), Queensland (23.8%), and Tasmania (23.8%). Teachers were aged between 29 and 62 years and had between 1 and 20 years of teaching experience. Class sizes ranged from 7 to 30. Most of the teachers had a Bachelor degree (57%). Ninety-five percent of the teachers had participated in in-service training (95%) and some of them (19%) taught composite classes—more than one grade level (e.g., Years 2/3). 

The results of this study show different coping strategies used by teachers to enhance their occupational well-being. These strategies were coded into ten themes and categorised under two headings, namely personal well-being-enabling initiatives (five themes) and school-based well-being-enabling initiatives (five themes) as presented below with illustrative quotes. 

### 3.1. Personal Well-Being Enabling Initiatives

***Theme 1—Setting boundaries:*** The teachers emphasised the importance of learning to set personal limits as a critical personal strategy for managing occupational well-being. They also highlighted the necessity of saying “no” to additional responsibilities when necessary to preserve personal time and mental health, and stressed the importance of prioritising one’s own well-being over external demands. For many teachers, this process developed over time through experience.


*“So, I think I’ve learned to set my boundaries. Now I’m older and better at saying, this is what I can do and take it or leave it sort of thing in a nice way. But I think leadership in every school I’ve worked at they’ve always tried to take more and push more for more. …So, I think it’s important to create boundaries and to say what you want to do and what you physically can’t do…”*
(Zelon)


*“I think for the most part I would say to myself, I can only do what I can do within the plan that I’ve got and I’m not going to be sitting here till 11 o’clock at night marking papers and doing whatever if I don’t get it done within a certain time, it just doesn’t get done. So, I just figured, if I don’t do this, or this or this, it’s not gonna matter so much… and that was just a survival thing because, I’ve got one friend who is a person who can’t leave those things…, and she had to take time away from teaching a couple of times…. She just wearied herself to the bone, and I’m not the type of person that will do that. I will just let you know that’s enough. And I’ll do what I think is important, regardless of what anyone else is telling me is important. I’ll just make up my own mind and decide if I can do that, or if I can’t, and if I can’t, I just leave it”*
(Lally)

Similarly, teachers reported setting boundaries by restricting work-related activities, such as only responding to emails during school hours. This discipline helped them preserve personal time.


*“Coping is sometimes saying ‘no’ to certain things. You know, being able to stop, but also standing your ground also to certain things …, like emails from parents. And I’ll say right now I don’t respond to emails at all. So that parents know that I also have a life outside of school. …, I’m going to respond in the morning and all that. …, I really disciplined myself to respond like during school time. So that it doesn’t become a habit for parents to just think they can just email me any time. Really discipline myself to just stop, and enjoy something else.…”*
(Saks)

***Theme 2—Exercise and physical health*:** Teachers found that exercise and physical health practices, such as walking, running, or going to the gym and maintaining a healthy diet, were effective ways to unwind and recharge after demanding days in the classroom. These routines provide teachers with a sense of control over their mental well-being, enabling them to manage stress more effectively.


*“I exercise 3 to 4 times a week, and yeah. Eat well, try to just eat well and exercise, and going outside when I can. Take a walk. Enjoying nature and stuff that still helps me with my well-being. Going to the seaside. and that sort of stuff”*
(Mira)


*“Every five weeks I’m taking a mental health day, and I’m going away for the weekend. And that’s a big improvement than what I’ve done previously. I’m taking some time out in the holidays to go away and look at the ocean”*
(Ash)


*“Going for a walk, Pilates’, talking to colleagues, talking to friends…. I suppose smiling when you meet someone and genuinely interested. Finding something that is common ground. I suppose just going and talking to them”*
(Maz)

***Theme 3—Social support and interactions:*** Spending time with friends enhanced teachers’ occupational well-being, providing a reliable source of emotional relief and balance outside the school environment. Teachers underscored the importance of friendships in offering a space to unwind, share personal challenges, and maintain a healthy work–life balance.


*“So, getting out, camping and spending time with friends. Like everyone, kind of worked hard. But then they are really fun”*
(Sey)


*“I suppose getting a good network of friends. Make time for doing things in the evening. So, I catch up with friends in the evening sometimes. I do pilates with a friend once a week so I have to go with her, and I catch up with a friend regularly to walk her dog or dogs”*
(Maz)

Social interactions with friends also allowed teachers to disconnect from their work and engage in restorative activities.


*“I make time in the evenings or weekends to do something with my friends. It helps me relax and enjoy time away from school.”*
(Mira)

***Theme 4—Mental Health and Mindfulness practices:*** Many teachers emphasised the importance of actively addressing their mental health needs and adopting mindfulness practices to navigate the pressures of their profession. Some teachers highlighted the value of seeking professional help to manage their mental health. Regularly consulting a mental health professional allowed them to develop healthier coping mechanisms and gain perspective on the challenges of teaching.


*“I’m getting better at not internalising it. Like I used to think, particularly in the time when I wasn’t coping and ended up going on workers comp. I thought I should be able to deal with it myself. And I was keeping it to myself, then. So, it was just becoming big. I see a psychologist now. But I’ve only seen her twice in the last 12 months, because I did have a period where I was seeing her regularly recently in the last two years”*
(Ash)

In addition to professional support, mindfulness practices such as deep breathing and meditation were mentioned as practical tools for reducing stress.


*“I just take a deep breath, and I just pray to God. Am a Christian, I just pray to God to help me. That’s what I do to be honest. I go to the gym to exercise, so that helps me too mentally”*
(Zem)

***Theme 5—Work–life balance:*** Teachers maintained a balance between work and personal life by leaving work-related stress at school and by ensuring to relax.


*“It was that, when you leave school for the day, walk out the gate knowing that you did everything you could. You worked as hard as you could, and then you leave it there. You don’t take it home with you. So, if I have had a bad day, or the class I’m teaching is challenging, then I don’t take that home with me and think about it all night at home. I put it down there at work and then I tackle the next challenge the next day when I come. I think it’s important to have that work/life balance. let everything that has happened to remain in school so you don’t go home with them”*
(Raba)

They highlighted the role of personal activities in maintaining work–life balance. By consciously separating their professional duties from their personal time, teachers were able to create space for rest and relaxation. Others chose to work part-time and shared that this flexibility was key to managing workload while maintaining a healthy balance.


*“I try to relax on the weekends. I’m doing gardening, and I have a pool which is really nice so I’m lucky. We go down to the beach and go swimming. So, you know, with my children, and I’ve got 2 at home, and 2 stepchildren as well, that are here at weekends. So, you know, we take them off to the beach, and things”*
(Esan)


*“Only because I work point 8. If I was full time, I wouldn’t have been able to do it. Point 8 is 4 days a week. One day off. Oh, on that one day I would do my school work. May be 4 or 6 h of school work. So, that I can spend my weekends more free with my family but also when it gets to Sunday evening I am also doing school work”*
(Maz)

### 3.2. School-Based Well-Being Enabling Initiatives

***Theme 1—Supportive Leadership***: Teachers reported supportive leadership as an important strategy to foster their occupational well-being, highlighting the critical role that school leaders play in creating an environment where they [teachers] feel valued, heard, and supported. They highlighted the importance of open communication with leadership. They also underscored the positive impact of having a leadership team that is aware of and responsive to teachers’ well-being needs.


*“I make sure I’m very open with my supervisors. So, if I’ve got things going on in my home world, I try and let them know what is going on. So, they will be more empathetic to my situation. If something was to happen or that kind of thing”*
(Raba)


*“I trust and I know that my assistant principal and my principal are very aware of my personal well-being needs. And I get the support that I need to deal with those”*
(Ash)

Leadership that actively provides support was seen as crucial for teachers’ success. Teachers appreciated it especially when the principal valued and protected staff.


*“I think I felt more valued then, I felt more supported. I think the principal that I had then was the type of person that is always, if a parent complained about something, she’d pass it on to you if she didn’t think it was warranted. Whereas the principal that I have [now]…, they’re more likely to listen to what the parent had to say and make you more accountable to the parent, follow-up phone calls, those sorts of things all the time you know. That principal I had first had a psychology degree, was able to make staff feel valued and part of the school community, and she would protect you”*
(Lally)

***Theme 2—S**upport from colleagues:*** Teachers emphasised the importance of connecting with colleagues, sharing experiences, and building a sense of community within the school environment. The ability to rely on peers for emotional and practical support fostered a sense of community within the school, helping teachers manage the pressures of their work.


*“Talking with colleagues after school, you have a really hard day you go to the staff room and you debrief with your colleagues. They usually know what I’m talking about”*
(Sod)


*“I think it is also important to get into the staffroom and interact with other people. It kind of even limits any stress or kind of just brings you back into perspective. So that you don’t get stuck in your classroom doing everything you need to do. Sometimes it’s important to prioritise just being social, interacting with other people so that’s helpful. And then as well I got given some advice when I started my teaching career”*
(Raba)

Having a supportive and collaborative workplace where colleagues mutually respect and support each order contributed significantly to a positive work environment. For newer teachers, the support from colleagues was invaluable.


*“I have respect for my colleagues, and I know that they have respect for me. They have a lot of respect for me and my role. We have bond together… which is important. You can be yourself. Yeah, everybody can be (themselves)…I would say that 90% of the staff feel like they can be themselves and not be judged”*
(Ash)


*“Talking to colleagues actually, and especially because I was, quite new to teaching, and I had a really good sort of bunch of colleagues around me, and I could talk to them about any problems at work, and they would support me”*
(Esan)

***Theme 3—Flexibility and autonomy:*** Teachers pointed out the importance of flexibility in managing additional tasks.


*“If genuinely teachers are expected to do things over and above the classroom role like paperwork, for example reports, personalised plans, stuff like that, they’re generally given time out of class to work on those”*
(Ash)

Having autonomy in how and when work is conducted, especially in alternative (such as part-time and online distance education) teaching settings, empowered teachers to perform their duties more effectively while maintaining personal well-being. The teachers shared how autonomy over workload and external interests helped them manage stress.


*“Because I am a part time teacher. I can maintain my stress okay. I feel like it’s in hand because I have lot of interest outside of school with animals or community, volunteering, whatever, and I have flexibility in my role. I feel a balance, and I feel happy with that. So, I can maintain any of the stress build-up from school. I can let it go. I can release it. I’m feeling good about my well-being right now”*
(Nath)


*“I like distance [online] education because it allows for a lot more flexibility. I’m just happy with where we are at the moment.”*
(Penny)

***Theme 4—Resource availability:*** Teachers highlighted the importance of having the tools they need to effectively manage their classrooms and deliver quality instruction. They acknowledged the challenges that arise when resources are insufficient.


*“As far as the school providing me with the resources I need to work with the kids, I don’t think it’s a realistic expectation. Schools would probably need double the funding they have now to do that”*
(Ash)

Despite these limitations, teachers often find ways to make the best use of the resources available, but the lack of adequate support can contribute to added pressure and stress.


*“If I make a resource, I keep it. I don’t throw it out. I laminate it and I keep it, so it’s always there. But you’re always creating new resources too. I’ve got a lot of stuff online that I’ve made over the years and then I can go back and reuse it or I can tweak it. I try to work smarter not harder”*
(Ash)

Having enough resources impacts not only the effectiveness of teaching but also the overall classroom environment. When these resources are lacking, it becomes much harder for teachers to meet the expectations placed upon them.


*“It’s not necessarily about the training it could be the physical environment, and it could be about the support that teachers need in order to be able to cater for the different needs. We know what the needs are. We understand what they are. We got that training…. We’ve got all tht. We just don’t have the means to actually carry out a program that would support the situation”*
(Lally)

***Theme 5—Proactive approaches to address challenges:*** This involves anticipating potential issues—whether related to classroom management, student behaviour, or workload—and implementing strategies to manage them effectively. Teachers emphasised the importance of taking proactive steps to maintain control over their professional environment. The teachers provided examples of proactive strategies used with students to prevent disruptions.


*“We tried things like proactive breaks, where the student knows, they are going to have a break at a set time, he calls it the ‘Fun House’, so it’s where he goes and they set up the punching bag, different activities. So, we’re not waiting for a meltdown to happen, then [before] taking him out for break, No!”*
(Saks)

They also emphasised the importance of individualising instruction based on student strengths as a proactive measure.


*“Getting to know the kids through the parents. Having meetings around what their strengths are…, manipulating the curriculum a little bit to be able to accommodate those strengths. Working around the things that might be preventing them from learning.… So really just individualised, tailored, targeted instruction”*
(Wet)

### 3.3. The SHIELD Model to Enhance Teachers’ Occupational Well-Being

Based on the findings and identified themes from the qualitative exploration of Australian primary school teachers’ experiences in relation to their occupational well-being and coping strategies, a comprehensive teacher support model was developed as a major outcome of this study. This model, named SHIELD, combines both personal and school-based coping strategies to provide a holistic approach to teacher well-being and retention in the profession (Figure 1). By implementing these comprehensive strategies, a sustainable and supportive work environment can be created for teachers. Teachers can protect their well-being with the SHIELD model. The SHIELD model is a comprehensive framework designed to enhance teacher well-being and effectiveness by integrating personal and school-based coping strategies. The acronym SHIELD stands for

**S**upport from Colleagues: The foundation of the SHIELD model is robust support from both colleagues and leadership. In a teaching environment, mutual support and collaboration among colleagues create a sense of community and shared responsibility. Teachers benefit from having colleagues who understand their challenges and provide emotional and practical support. This fosters an inclusive and supportive school culture. Ash emphasised, “The people I work with, they’re really good. They are supportive, everybody works together, it’s a collective responsibility for the children”.**H**ealthy Lifestyle and Exercise: Maintaining a healthy lifestyle is vital for managing stress and enhancing overall well-being. Regular exercise, a balanced diet, and sufficient rest contribute to physical and mental health, enabling teachers to cope better with the demands of their profession. Encouraging teachers to prioritise their health can improve energy levels, reduced stress, and greater resilience. Mira shared, “I exercise 3 to 4 times a week, and eat well, try to just eat well and exercise, and going outside when I can”.**I**nteractions and Social Connections: Social interactions and connections within and outside the school environment are essential for alleviating stress and maintaining a positive outlook. Engaging with colleagues and friends, participating in social activities, and building strong relationships can provide emotional support and a sense of belonging. These interactions help teachers to debrief, share experiences, and gain new perspectives on handling challenges. Raba noted, “It is important to get into the staffroom and interact with other people. It kind of even limits any stress or kind of just brings you back into perspective”.**E**mpathy and Understanding in the Workplace: Empathy and understanding from both colleagues and leadership are critical components of the SHIELD model. When teachers feel understood and supported, they are more likely to express their concerns and seek help when needed. Empathy in the workplace fosters a culture of care and respect, which can significantly reduce feelings of isolation and stress. Sore mentioned, “They (leadership) have that bond, everybody does. They are pretty good in terms of support for our mental health”.**L**eadership that Listens: Effective leadership that listens and responds to teachers’ needs is fundamental for a supportive work environment. Leaders who are approachable, empathetic, and proactive in addressing teacher concerns can significantly enhance teacher morale and job satisfaction. Providing opportunities for professional development, recognising achievements, and involving teachers in decision-making processes are ways leaders can demonstrate their commitment to teacher well-being. Open communication channels between teachers and school leaders ensure that teachers feel heard, valued, and supported in their professional journey. Penny stated, “I’ve had really good support and understanding from them [leadership]. Goes both ways as well. Being honest and upfront”.**D**evelopment of Personal and Professional Boundaries: Setting and maintaining personal and professional boundaries is crucial for preventing burnout and ensuring sustainable well-being. Teachers need to learn how to manage their workload effectively, prioritise tasks, and say no when necessary. Encouraging teachers to establish clear boundaries helps them balance their work and personal lives, reducing the risk of stress and exhaustion. Zelon said, “I think I’ve learned to set my boundaries. Now I’m older and better at saying, this is what I can do, and take it or leave it sort of thing in a nice way”.

**Figure 1 behavsci-14-00918-f001:**
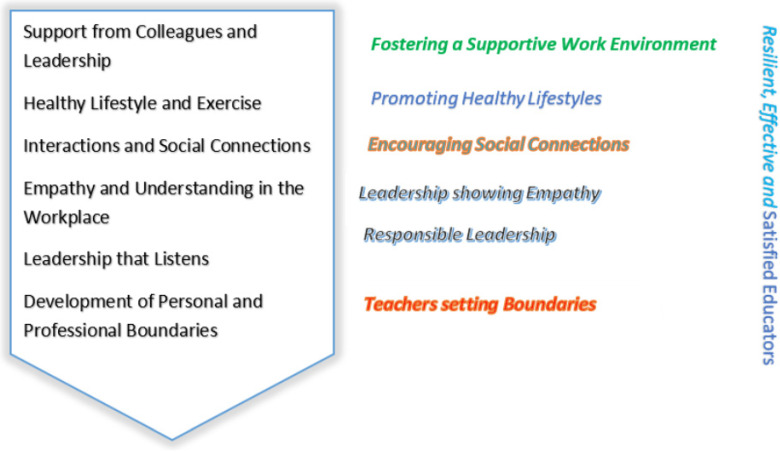
The SHIELD model.

The SHIELD model provides a holistic approach to supporting teacher well-being by integrating key elements that address both personal and professional needs. By fostering a supportive environment, promoting healthy lifestyles, encouraging social connections, practicing empathy, providing responsive leadership, and helping teachers establish boundaries, the SHIELD model aims to create resilient, effective, and satisfied educators. Implementing this model can lead to improved teacher retention, enhanced teaching quality, and better educational outcomes for students.

## 4. Discussion

This study was conducted to explore the occupational well-being-enabling coping strategies employed by Australian primary school teachers. Teaching is a demanding profession, both physically and mentally, requiring high emotional involvement, which can lead to significant anxiety and stress [57,58]. As teacher coping methods are increasingly recognised among the key factors determining teacher effectiveness [5], an understanding of these strategies with a view to improving them is essential. Effective coping strategies aim to help teachers maintain personal integrity and a sense of control over their challenges [59,60]. Coping strategies can influence how stress is experienced and may reduce its impact [61,62]. Given that teaching is inherently an emotional endeavour [63], coping is essential for sustaining teacher well-being and preventing burnout [43].

In this study, teachers’ coping strategies were categorised into two broad areas, personal and school-based well-being initiatives, which were synthesised into the SHIELD model. The SHIELD model aligns closely with the OECD teacher occupational well-being framework, which focuses on cognitive, subjective, physical, mental, and social well-being. The SHIELD components, such as Support, Empathy, and Interaction, enhance social well-being by fostering collaboration and emotional support, while Healthy Lifestyle and Leadership that Listens promote physical, mental, and subjective well-being by encouraging stress management, resilience, and job satisfaction. By integrating personal and school-based coping strategies, SHIELD also addresses cognitive well-being, helping teachers manage workloads and set boundaries to prevent burnout. Together, these elements create a holistic approach to improving teacher well-being.

The findings suggest that teachers exhibit distinct patterns of coping based on their individual strengths and skills [59]. Different strategies work for different teachers, depending on their personal circumstances and the support available to them. A significant finding from this study is the crucial role of support from both colleagues and leadership in managing stress. Teachers reported that their work environments were generally supportive, with school leadership providing the necessary resources and feedback to help them perform their roles effectively. Social support, including emotional support like sharing concerns with colleagues, was a key coping strategy, aligning with previous research [64,65,66,67]. Seeking advice from leadership also played a crucial role in teachers’ positive occupational well-being. Supportive leadership is essential as a stressful work environment can negatively impact teachers’ self-efficacy, job satisfaction, and emotional well-being [24,25,68,69]. The literature strongly supports the importance of principal support in enhancing teacher well-being [70,71,72].

Social connections, including strong relationships with children, colleagues, and parents, were vital for coping. Good rapport with all stakeholders improved teachers’ occupational well-being and helped them manage the demands of their profession. Teachers emphasised the importance of professional relationships as a key coping strategy, as highlighted in previous studies [60,66]. Additionally, maintaining a balanced lifestyle, including ensuring time for lunch breaks and good sleep, was crucial for managing stress effectively. Teachers also noted that self-efficacy in managing classroom dynamics and parent relationships significantly enhanced their well-being, supporting findings from Kilgallon et al. [73] and Kennedy et al. [74].

Empathy and understanding from leadership were essential in supporting teachers’ well-being. Teachers appreciated when leadership allowed them to reduce their workload or adjust their roles, such as switching to part-time work or changing from classroom teaching to support roles [75]. This flexibility helped teachers maintain their stress levels and occupational well-being, which aligns with research suggesting that manageable workloads protect against stress [71,72]. Empathetic leadership that provided time for teachers to complete their work, recognised their efforts, and created a supportive work environment was shown to enhance job satisfaction and well-being, echoing findings by Skinner et al. [76] and Walter and Fox [32].

Effective leadership that listens and responds to teachers’ needs is critical for maintaining teacher well-being. Teachers who felt supported by their leadership were more likely to seek help, ask questions, and suggest improvements, such as establishing proactive breaks for students with special needs. This proactive approach helps manage classroom dynamics and supports teacher well-being, particularly in diverse and challenging classroom settings [24]. Additionally, teachers who felt appreciated and valued by their leadership were better able to manage stress and maintain a positive outlook, further emphasising the importance of strong, supportive leadership in schools [77].

Teachers used a variety of personal coping strategies to manage stress, including exercise, healthy eating, nature walks, and engaging in religious practices such as prayers. Exercise was particularly highlighted as an effective way to support occupational well-being, corroborating previous findings [64]. Faith and prayer also served as significant coping mechanisms, with teachers turning to religion during stressful times, consistent with the literature [5]. Setting and maintaining personal and professional boundaries was another critical coping strategy. Teachers reported that experience had taught them to set clear limits on their workload and to prioritise tasks, leaving less critical work for another time. This ability to say “no” to non-essential tasks and manage time effectively was crucial in preventing burnout and maintaining a healthy work–life balance, consistent with findings by Lindqvist et al. [60].

### 4.1. Implications for Practice

The findings from this study have significant practical implications for enhancing teacher well-being and retention. Teachers employ a variety of coping strategies to manage the stress inherent in their profession, and the support they receive from school leadership plays a critical role in their overall occupational well-being. Leadership support is paramount in reducing the impact of stress on teachers. Principals and school leaders should be intentional about providing robust support systems and professional development opportunities. Recent studies have reported that the most effective well-being measures are those integrated into supportive whole-school cultures that reduce burdensome workloads while enhancing feelings of autonomy, relatedness, and competence. Generally, teachers prefer school policies and practices that promote manageable workloads over one-off or short-term well-being activities [78].

Additionally, the study revealed that establishing boundaries is a key coping strategy for teachers. This suggests that schools should incorporate boundary-setting into professional development programs, offering teachers the support they need to maintain a healthy work–life balance [60]. Furthermore, teachers reported seeking professional help as part of their coping strategies, indicating a potential need for schools to establish counselling centres specifically for teachers, in addition to those available for students. In-service training should also focus on expanding teachers’ coping repertoires to help prevent stress and burnout in their current roles [79].

### 4.2. Strengths and Limitations of This Study

This study’s qualitative approach allowed for an in-depth exploration of teachers’ personal experiences and coping strategies, providing rich, detailed insights into the specific ways teachers manage stress and maintain well-being. The development of the SHIELD model is another notable strength, providing a structured, practical framework that can be directly applied in educational settings to enhance teacher well-being and retention. However, this study was limited to a specific group of Australian primary school teachers, which may not be representative of all teachers. The findings may not be transferable to teachers in different educational contexts or geographical regions. Additionally, the self-reported data may have introduced social desirability bias, as participants may have presented themselves in a more favourable light or may not have accurately recalled their coping strategies and experiences.

## 5. Conclusions

This study identified key personal and school-based initiatives that teachers use to cope with job demands. Personal strategies include setting boundaries, maintaining a healthy lifestyle, seeking social support, and practicing mindfulness. School-based initiatives, such as supportive leadership, collaboration, flexibility, and professional development, enhance well-being. Together, these approaches help teachers manage stress, prevent burnout, and maintain a healthy work–life balance. The SHIELD model integrates these strategies into a practical framework, offering a comprehensive approach to fostering teacher resilience and well-being.

The SHIELD model emphasises the need for robust support systems, healthy lifestyle promotion, social connection, empathetic leadership, and clear personal and professional boundaries. By implementing these strategies, schools can create a sustainable and supportive environment that not only enhances teacher well-being but also improves retention rates and educational outcomes. Future research could adopt a longitudinal approach to track changes in teacher coping strategies and well-being over time. This would provide insights into how these strategies evolve with experience and changing professional demands. Additionally, the long-term effectiveness of the SHIELD model and its applicability across different educational contexts could be explored.

## Data Availability

The dataset is available upon reasonable request from the corresponding author (J.C.N.).

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
