# Peer review of "“SHIELDing” Our Educators: Comprehensive Coping Strategies for Teacher Occupational Well-Being"

_behavsci, 2024, doi:10.3390/bs14100918_

Round 1

Reviewer 1 Report

Comments and Suggestions for Authors

Thank you for the opportunity to review this interesting and valuable article reporting on a study which investigated personal and school-based well-being initiatives that teachers use for maintaining their occupational well-being, and the development a coping strategy model that enhances teachers' occupational well-being.

Introduction

·         Uses current literature to set the context and significance of the study (teacher stress, burnout and attrition).

·         Aim of this study is included

·         You have not discussed the wellbeing framework which you are applying in this study. However, I would suggest that the focus of your study is actually “teacher coping” rather than “teacher wellbeing” There is a lot of current literature in the field of teacher wellbeing which you have not considered in this article so, I would recommend that you remove references to teacher wellbeing and focus on teacher coping.

Method

·         Project design, sampling method, data collection, data analysis were explained clearly.

·         Good number of participants

·         Need to include research question for this study.

Results

·         Figure 1 “showing most of the teachers were female” (line 174 on page 4) appears to be missing from the article.

·         Table 1 clearly presented themes and relevant teacher quotes

·         Some spelling mistakes in the teacher quotes e.g. Pilates and principal are spelt incorrectly.

·         As there are elements of wellbeing not addressed in the SHIELD model, perhaps consider amending throughout the article to specify that SHIELD model is a model for supporting teacher coping (remove reference to wellbeing).

Discussion

·         Sound discussion of teacher coping findings in relation to current literature

·         Again here, as you have not included a wellbeing framework in the discussions, please consider amending to specify that the SHIELD model is a model for supporting teacher coping (rather than wellbeing) in the stressful school environment.

·         Limitations have been acknowledged

Conclusion

·         Please clearly answer the research question in the conclusion (hard to determine if this has been done as research question not included in article).

·         Recommendations for further research in this field have been included

Recommendation: Minor Revisions

Comments on the Quality of English Language

Acceptable

Author Response

Reviewer: 1

Thank you for the opportunity to review this interesting and valuable article reporting on a study which investigated personal and school-based well-being initiatives that teachers use for maintaining their occupational well-being, and the development a coping strategy model that enhances teachers' occupational well-being.

Response: We thank the reviewer for their kind words.

Introduction

  • Uses current literature to set the context and significance of the study (teacher stress, burnout and attrition).

Response: We thank the reviewer for the comment.

 Aim of this study is included

Response: We thank the reviewer for the comment.

  • You have not discussed the wellbeing framework which you are applying in this study.

Response: We have included the framework used in the methods section. Please see Lines 130-133:

This study is part of a bigger project that investigated Australian primary school teachers’ occupational well-being that was guided by the OECD teacher well-being framework, which includes four core wellbeing dimensions (cognitive, subjective, physical/mental and social) [16].

However, I would suggest that the focus of your study is actually “teacher coping” rather than “teacher wellbeing” There is a lot of current literature in the field of teacher wellbeing which you have not considered in this article so, I would recommend that you remove references to teacher wellbeing and focus on teacher coping.

Response: We thank the reviewer for the comment. However, as already mentioned, this study is part of a larger study on teacher occupational well-being and the OECD framework was used as a guide in the development of the study. Additionally, as indicated in the manuscript, the study aimed to investigate the personal and school-based well-being enabling initiatives that teachers use to maintain their occupational well-being. We therefore prefer to retain the information on teacher wellbeing. Nonetheless, we have included coping in the title of the paper. It now reads thus:

“SHIELDing” Our Educators: Comprehensive Coping Strategies for Teacher Occupational Well-Being.

Method

Project design, sampling method, data collection, data analysis were explained clearly.

  • Good number of participants
  • Need to include research question for this study.

Response: We thank the reviewer for the comment. We have now included the research quesstion in the last paragraph of the Introduction section. Please see lines 120-122.

Therefore, this study sought to answer the research question “What are the personal and school-based occupational well-being enabling initiatives that teachers utilise to cope with the demands of their job?”

Results   

   Figure 1 “showing most of the teachers were female” (line 174 on page 4) appears to be missing from the article.

Response: This was an error. Narrative about the demographic profile of the participants is presented at the begining of the Results section.  Please see Lines 200-209.

A total of 21 Australian primary school teachers comprising, classroom teachers (n = 11), specialist teachers (n = 4), learning support teachers (n = 2) and school leaders (n = 4) participated in this study. Most of the teachers were females (95.2%) and worked in public schools (57.1%). The participants were from five Australian states - Australian Capital Territory (9.5%), New South Wales (23.8%), Northern Territory (19%), Queensland (23.8%) and Tasmania (23.8%). Teachers were aged between 29 to 62 years and had between 1 to above 20 years of teaching experience. Class size ranged from 7 to 30. Most of the teachers had a Bachelor degree (57%). Ninety-five percent of the teachers had participated in in-service training (95%) and some of them (19%) taught composite classes - more than one grade level - (e.g., Years 2/3).

Table 1 clearly presented themes and relevant teacher quotes

  • Some spelling mistakes in the teacher quotes e.g. Pilates and principal are spelt incorrectly.

Response: We thank the reviewer for the feedback. The spelling mistakes have been corrected.

  • As there are elements of wellbeing not addressed in the SHIELD model, perhaps consider amending throughout the article to specify that SHIELD model is a model for supporting teacher coping (remove reference to wellbeing).

Response: We thank the reviewer for the feedback. As stated before, the project is centred on teacher occupational well-being, hence the reference to well-being. The aims of the study were two-fold: (1) To investigate the personal and school-based well-being enabling initiatives that teachers use to maintain their occupational well-being, and (2) to develop a coping strategy model that enhances teachers' occupational well-being.

The coping strategies are presented as effective initiatives to enable teachers maintain their occupational well-being and teaching quality.

Discussion

  • Sound discussion of teacher coping findings in relation to current literature

  Response: We thank the reviewer for the kind comment.

Again here, as you have not included a wellbeing framework in the discussions, please consider amending to specify that the SHIELD model is a model for supporting teacher coping (rather than wellbeing) in the stressful school environment.

  Response: We have included information in the Discussion section to demonstrate how the developed SHIELD model aligns with the OECD teacher occupational wellbeing framework. Please see Lines 527-535.

The SHIELD model aligns closely with the OECD teacher occupational well-being framework, which focuses on cognitive, subjective, physical, mental, and social well-being. The SHIELD components, such as Support, Empathy, and Interaction, enhance social well-being by fostering collaboration and emotional support, while Healthy Lifestyle and Leadership that Listens promote physical, mental, and subjective well-being by encouraging stress management, resilience, and job satisfaction. By integrating personal and school-based coping strategies, SHIELD also addresses cognitive well-being, helping teachers manage workloads and set boundaries to prevent burnout. Together, these elements create a holistic approach to improving teacher well-being.

  • Limitations have been acknowledged

 Response: We thank the reviewer for the feedback.

Conclusion

  • Please clearly answer the research question in the conclusion (hard to determine if this has been done as research question not included in article).

Response: We have included a statement at the begining of the Conclusion section that clearly answers the research question. Please see lines: 623-630.

This study identified key personal and school-based initiatives that teachers use to cope with job demands. Personal strategies include setting boundaries, maintaining a healthy lifestyle, seeking social support, and practicing mindfulness. School-based initia-tives, such as supportive leadership, collaboration, flexibility, and professional develop-ment, enhance well-being. Together, these approaches help teachers manage stress, pre-vent burnout, and maintain a healthy work-life balance. The SHIELD model integrates these strategies into a practical framework, offering a comprehensive approach to fostering teacher resilience and well-being.

  • Recommendations for further research in this field have been included

Recommendation: Minor Revisions

Response: We thank the reviewer for all their comments which have helped to improve the quality of our manuscript.

Reviewer 2 Report

Comments and Suggestions for Authors

Literature review

The literature review includes reference to relevant publications on teacher wellbeing/stress/burnout and makes a compelling argument for the present study. I did feel that sometimes the presentation of teachers' working lives was a little simplistic, e.g. teaching causes 'negative emotions' which lead to stress. The paper would be improved by discussing the possible causes of such emotions in more detail. I would recommend, for example, looking at Jane Perryman's recent work, as she discusses the disappointment experienced by teachers who enter the profession wanting to make a difference but feel unable to do so in the current context of high-stakes accountability. 

There is also a lack of historical context - the authors mention accountability, but do not explain how accountability has increased significantly for teachers as part of neoliberal pressures on education systems in the past 30-so years. I realise the paper is located within a psychological approach but I think just some minor historical contextualisation which provides a partial explanation for why teachers experience such high accountability demands would be useful in setting the scene. 

The literature review is also strangely decontextualised. Your data is exclusively Australian primary teachers, so I would expect some discussion around the Australian context to be present in this introductory section (see perhaps the work of Meghan Stacey, Anna Hogan, or Nicole Mockler?) as well as literature which addresses the specific situation of primary teachers. In short, this introduction/lit review should give us an overview of why Australian primary teachers are particularly worthy research subjects when exploring teacher stress.

Methods 

Again, it is unclear why primary teachers have been chosen. This should be explicitly stated in the methods, after being signposted in the literature review.

You state the sampling was purposive, but then describe sampling techniques that appear to be more like convenience sampling (e.g. snowballing). I think the section on sampling needs to address this issue more directly. It is very difficult to recruit teachers and readers will understand and sympathise with this, but transparency is necessary around how recruitment was conducted. Was it just that you recruited 21 primary teachers (you didn't set out to stratify/balance the sample according to leadership responsibilities, career stage, gender etc?) If s, this is fine, but I do think you have relied on an element of convenience sampling which needs to be acknowledged. It would also be useful to know if the sample were self-selecting. 

Results

Sometimes the illustrative quotes you choose do not seem to support the theme you have generated/identified. For example, you have a theme of 'Exercise and physical health' but then two of the quotes mention drinking wine. Similarly, the quotes about social interaction seem to confuse/conflate  colleagues and friends, and in Flexibility and Autonomy the quote about distance education doesn't seem to support the theme- the reason given is relationships rather than flexibility/autonomy. Perhaps you could consider providing some commentary on these themes, rather than just having a table of the theme and quotes? This would provide the reader with some insight as to why these quotes have been grouped together and classified in such a way.  I do wonder though if the analysis needs to be revised given these issues. 

In terms of the SHIELD model, three of the points in the model are around having supportive leadership (1, 4 and 5). This seemed to me to be repetitive - perhaps these need to be better differentiated?

Discussion

You state that 'This study was conducted to explore the coping strategies employed by Australian 262 primary school teachers, utilising the SHIELD model as a framework.' But wasn't this essentially a grounded theory  paper, with the framework developing as a result of the research? You don't analyse the data using the framework as a pre-existing model, but instead the SHIELD model is developed out of your research - yes? This needs clarification.

In the discussion you mention alcohol as a negative 'avoidant' coping strategy. This relies on psychological theory about coping strategies which has not been adequately discussed as a theoretical framework in the introduction/lit review and/or methods. Being more explicit about the psychological theories you are using to identify coping strategies could, I feel, improve your analysis and the rigour of your findings. 

In 4.1 you recommend 'stress management workshops.' I wonder if you might benefit from reading Brady & Wilson's brilliant article on teacher wellbeing initiatives in schools https://www.tandfonline.com/doi/full/10.1080/0305764X.2020.1775789 - this might just inform some of the recommendations you make and help you to present them in a more nuanced manner. 

In the appendix you include your semi-structured interview questions. I was a little concerned that the first was a leading question (tell me how well the school supports you?) In future research, I would suggest rewording such questions in a more neutral way e.g., tell me about ways in which teachers are supported in this setting...

Comments on the Quality of English Language

L83-84 please reword as this does not quite make sense

L99 should this be resilience strategies?

L160-161 This should be moved to the section on sampling/participant recruitment.

L172-181 This should be in the methods section rather than the results section.

Lally illustrative quote - wearied herself to the bone - should this be wore herself/weared herself to the bone? (or sic?)

Maz illustrative quote - palates, should this be pilates?

L211 - there seems to be a repeat of Interaction and social connection here?

Author Response

Reviewer 2 comments

Literature review

The literature review includes reference to relevant publications on teacher wellbeing/stress/burnout and makes a compelling argument for the present study. I did feel that sometimes the presentation of teachers' working lives was a little simplistic, e.g. teaching causes 'negative emotions' which lead to stress. The paper would be improved by discussing the possible causes of such emotions in more detail. I would recommend, for example, looking at Jane Perryman's recent work, as she discusses the disappointment experienced by teachers who enter the profession wanting to make a difference but feel unable to do so in the current context of high-stakes accountability. 

Response: We thank the reviewers for the comment. We have now provided a more compelling argument for our study. We have elaborated on and incorporated Jane Perryman’s recent work in our explanation of the connections between teaching, negative emotions and stress.

Please see Lines 60-66: Additionally, teachers face heightened accountability within the school context [18]. Teachers cite workload, poor work-life balance, and the target-driven culture shaped by government initiatives as key reasons for leaving. While many enter the profession for altruistic reasons—such as wanting to make a difference—the reality of teaching, shaped by accountability and performativity pressures, leads to disappointment and early exit [19]. This reflects a disillusionment with the broader teaching context and the pressures of accountability.

Please see Lines 71-80: Literature reports that teaching daily poses numerous challenges, causing teachers to experience stressful events and negative emotions like anxiety or anger during class [26]. Most especially for primary school teachers who have reported more stress in literature than other teacher type [27,28]. A recent literature review [29] clearly indicated that pri-mary school teachers were the most stressed and may face more challenges in managing disruptive behaviour from young children, which can negatively impact their well-being [30,31]. They may require more energy and patience to handle younger children, which can be exhausting and require a high degree of emotional labour and experience higher levels of workload and stress due to the constant need for attention and supervision of young children [32]. The perception of teaching as stressful may be influenced by coping responses and social support [33].

There is also a lack of historical context - the authors mention accountability, but do not explain how accountability has increased significantly for teachers as part of neoliberal pressures on education systems in the past 30-so years. I realise the paper is located within a psychological approach but I think just some minor historical contextualisation which provides a partial explanation for why teachers experience such high accountability demands would be useful in setting the scene. 

Response: We have now provided some historical context in the introduction. Please see lines 44-50.

High levels of teacher occupational stress have been documented globally [3,4,6] and within Australia [3,8], which is the focus of this study. The findings suggest that administrative tasks are a greater source of stress for teachers than classroom teaching hours. Australian teachers, like their international counterparts, spend an average of 19 hours teach-ing, 7 hours planning, and 5 hours marking each week [3,8]. However, Australian teachers work an average of 43 hours per week, which is 5 hours more than the global average [8,3,9].

The literature review is also strangely decontextualised. Your data is exclusively Australian primary teachers, so I would expect some discussion around the Australian context to be present in this introductory section (see perhaps the work of Meghan Stacey, Anna Hogan, or Nicole Mockler?) as well as literature which addresses the specific situation of primary teachers. In short, this introduction/lit review should give us an overview of why Australian primary teachers are particularly worthy research subjects when exploring teacher stress.

Response: As indicated in the responses above, we have made the Australian context and justification for focus on primary school teachers more explicit.

Methods 

Again, it is unclear why primary teachers have been chosen. This should be explicitly stated in the methods, after being signposted in the literature review.

Response: The reason for choosing mainstream primary school teachers has now been added to the study methods.

Please see lines: 130- 133: This study is part of a bigger project that investigated Australian primary school teachers’ occupational well-being that was guided by the OECD teacher well-being framework, which includes four core wellbeing dimensions (cognitive, subjective, physical/mental and social) [16].

Please see lines: 142- 149: The focus on primary school teachers is justified by the unique challenges they face, including managing young students with diverse developmental needs, creating foundational learning experiences, and balancing heavy workloads with limited resources [23-25]. Primary school teachers often play a critical role in shaping children's early educational experiences, making their well-being essential for effective teaching and long-term student success. Additionally, primary educators are particularly vulnerable to stress and burnout, which underscores the importance of identifying strategies that support their occupational well-being [27-29].

You state the sampling was purposive, but then describe sampling techniques that appear to be more like convenience sampling (e.g. snowballing). I think the section on sampling needs to address this issue more directly. It is very difficult to recruit teachers and readers will understand and sympathise with this, but transparency is necessary around how recruitment was conducted. Was it just that you recruited 21 primary teachers (you didn't set out to stratify/balance the sample according to leadership responsibilities, career stage, gender etc?) If so, this is fine, but I do think you have relied on an element of convenience sampling which needs to be acknowledged. It would also be useful to know if the sample were self-selecting. 

Response: The research used purposive sampling to recruit the participants, where in only registered Australian primary school teachers and school administrators were invited to participate in the study. However, we agree with the reviewer that some element of convenient sampling was utilised as some of the interviewees were asked to invite their colleagues to participate in the study (snowballing technique). We have revised information about the sampling technique.

Please see lines 151 – 156.  A combination of purposive and convenience sampling techniques was used to recruit participants for this study. Initially, participants were recruited using purposive sampling, targeted at registered Australian primary school teachers and school leaders across Australia. Recruitment was conducted through educational forums and social media platforms dedicated to educators. Subsequently, snowballing was utilised to increase participant numbers.

Results

Sometimes the illustrative quotes you choose do not seem to support the theme you have generated/identified. For example, you have a theme of 'Exercise and physical health' but then two of the quotes mention drinking wine. Similarly, the quotes about social interaction seem to confuse/conflate colleagues and friends, and in Flexibility and Autonomy the quote about distance education doesn't seem to support the theme- the reason given is relationships rather than flexibility/autonomy. Perhaps you could consider providing some commentary on these themes, rather than just having a table of the theme and quotes? This would provide the reader with some insight as to why these quotes have been grouped together and classified in such a way.  I do wonder though if the analysis needs to be revised given these issues. 

Response: As suggested by the reviewer, we have now presented the themes and relevant quotes in the text rather than in a table and provided commentary on each theme. This has now provided a logical flow of the content.  Please see lines 214-440.

In terms of the SHIELD model, three of the points in the model are around having supportive leadership (1, 4 and 5). This seemed to me to be repetitive - perhaps these need to be better differentiated?

Response: Thanks for the comment. These points have now been differentiated. Point 1 now focuses on collaboration with colleagues only, point 4 is about empathy and understanding from colleagues and the leaders, while point 5 is targeted at leadership that listens. Please see Lines 454, 478 and 486.

Discussion

You state that 'This study was conducted to explore the coping strategies employed by Australian 262 primary school teachers, utilising the SHIELD model as a framework.' But wasn't this essentially a grounded theory paper, with the framework developing as a result of the research? You don't analyse the data using the framework as a pre-existing model, but instead the SHIELD model is developed out of your research - yes? This needs clarification.

Response: Yes, the SHIELD model was developed out of our research. The statement quoted above was an error and it has been corrected – refernce to the SHIELD mdel as a framework utilised for the study has been deleted.

In the discussion you mention alcohol as a negative 'avoidant' coping strategy. This relies on psychological theory about coping strategies which has not been adequately discussed as a theoretical framework in the introduction/lit review and/or methods. Being more explicit about the psychological theories you are using to identify coping strategies could, I feel, improve your analysis and the rigour of your findings. 

Response: This sentence has been deleted as it is no longer relevant.

In 4.1 you recommend 'stress management workshops.' I wonder if you might benefit from reading Brady & Wilson's brilliant article on teacher wellbeing initiatives in schools https://www.tandfonline.com/doi/full/10.1080/0305764X.2020.1775789 - this might just inform some of the recommendations you make and help you to present them in a more nuanced manner. 

Response: We thank the reviewer for the observation and suggestion. We have read the article and incorporated the important findings presented in the article to ensure that our recommendations are more evidence-based. Please see Lines 594-598

Recent studies have reported that the most effective well-being measures are those integrated into supportive whole-school cultures that reduce burdensome workloads while enhancing feelings of autonomy, relatedness, and competence. Generally, teachers prefer school policies and practices that promote manageable workloads over one-off or short-term well-being activities. [78].

In the appendix you include your semi-structured interview questions. I was a little concerned that the first was a leading question (tell me how well the school supports you?) In future research, I would suggest rewording such questions in a more neutral way e.g., tell me about ways in which teachers are supported in this setting...

Response: We thank the reviewer for the suggestion, which will be incorporated into future practice.

Comments on the Quality of English Language

L83-84 please reword as this does not quite make sense

Response: We have now reworded it. Please see lines: 97-100.

Bermejo-Toro et al. [7], Kim et al. [40] and Nwoko et al. [29] all reported that coping strate-gies help to improve teacher well-being and sustain teachers when they experience inadequate support [41,42], preventing burnout [43].

L99 should this be resilience strategies?

Response: This has been changed to resilience strategies.

L160-161 This should be moved to the section on sampling/participant recruitment.

Response: We do not agree with the reviewer on this point. This information is part of the study results and so we have left it in the results section.

L172-181 This should be in the methods section rather than the results section.

Response: Thank you for the observation. However, we prefer to keep it in the results section.

Lally illustrative quote - wearied herself to the bone - should this be wore herself/weared herself to the bone? (or sic?)

Response: The phrase “just wearied herself to the bone" is correct. It means someone has exhausted themselves completely, both physically and mentally, through intense effort or work. It's a figurative expression used to describe extreme fatigue or overwork.

Maz illustrative quote - palates, should this be pilates?

Response: We thank you for the comment. The word has now been spelt correctly.

L211 - there seems to be a repeat of Interaction and social connection here?

Response: We thank you for the comment. The repeated phrase has been removed.

We thank the reviewer for all their comments which have helped to improve the quality of our manuscript.
